# Assessing the Effectiveness of Modified Tibial Plateau Leveling Osteotomy Plates for Treating Cranial Cruciate Ligament Rupture and Medial Patellar Luxation in Small-Breed Dogs

**DOI:** 10.3390/ani14131937

**Published:** 2024-06-30

**Authors:** Eunbin Jeong, Youngjin Jeon, Taewan Kim, Dongbin Lee, Yoonho Roh

**Affiliations:** 1Department of Veterinary Surgery, College of Veterinary Medicine, Gyeongsang National University, Jinju 52828, Republic of Korea; dmsqls1196@gmail.com (E.J.); abcball@naver.com (T.K.); dlee@gnu.ac.kr (D.L.); 2Department of Veterinary Surgery, College of Veterinary Medicine, Chungnam National University, Daejeon 34134, Republic of Korea; orangee0115@gmail.com

**Keywords:** cranial cruciate ligament rupture, medial patellar luxation, tibial plateau leveling osteotomy, tibial tuberosity transposition, modified tibial plateau leveling osteotomy, dogs

## Abstract

**Simple Summary:**

In small-breed dogs, medial patellar luxation and cranial cruciate ligament rupture often occur concurrently in the adult dog, presenting through diverse mechanisms. Numerous methods have been developed to treat these conditions together. The combined osteotomy method may elevate the risk of joint instability and fractures. Therefore, a pre-contoured modified tibial plateau leveling osteotomy plate was designed to achieve the dual effect of tibial tuberosity transposition, a surgery performed in kneecap dislocation correction, alongside reducing the tibial plateau angle. In this cadaveric study, we examined the efficacy and stability of this modified plate. The findings indicate that this pre-contoured plate effectively prevents forward displacement of the tibia (shinbone) and medial (inner) dislocation of the kneecap while maintaining stability.

**Abstract:**

In small-breed dogs with concurrent cranial cruciate ligament rupture (CCLR) and medial patellar luxation (MPL), correcting both disorders is are essential for restoring normal gait. However, the previously described surgical treatment, using two osteotomy technique, poses a high risk of fracture and instability. Addressing CCLR and MPL with a single osteotomy and implant was considered superior to the conventional method. Therefore, a pre-contoured modified tibial plateau leveling osteotomy (PCM–TPLO) plate facilitating medial shifting of the proximal tibia was developed. We compared postoperative alignment and strength between this novel plate group and a conventional tibial plateau leveling osteotomy (TPLO) plate group using eight small-breed dog cadavers each. Additionally, we investigated the potential of the novel plate as an alternative to tibial tuberosity transposition. Postoperative alignment and strength were assessed through radiographs and mechanical testing. Measurements including tibial plateau angle, mechanical medial proximal tibial angle, and number of screws within the joint were also analyzed. There were no significant differences in all measured parameters. For the novel plate, the medial displacement ratio of the proximal tibia was confirmed to be approximately 30%, and the result was thought to be appropriate. These findings suggest that the PCM–TPLO plate could be a promising alternative for treating concurrent CCLR and MPL in small-breed dogs.

## 1. Introduction

Cranial cruciate ligament rupture (CCLR) and medial patellar luxation (MPL) are widely recognized as prevalent causes of hindlimb lameness in dogs [1,2]. While MPL and CCLR can manifest independently, their concurrent occurrence has been observed in 6–25% of cases in small-breed dogs, with older dogs being more susceptible [3,4,5,6]. Although the exact etiology of concurrent MPL and CCLR remains unclear, a wide range of anatomical and biomechanical factors are considered to play a role in their occurrence [1,4,7]. The pathogenesis of MPL is often characterized by an aberrant arrangement of the muscles responsible for the quadriceps mechanism and bone deformities of the femur and tibia, leading to persistent joint instability and abnormal tibial movement [8,9]. This, in turn, may culminate in increased stress on the ligaments, potentially resulting in cruciate ligament tears and laxity [4,10,11]. Conversely, partial damage to the cruciate ligament, resulting in joint instability, may induce joint laxity and subsequently contribute to the development of MPL [3]. A thorough understanding of the onset and impact of these diseases is crucial for effective surgical intervention [12].

Various surgical procedures have been developed for dogs with concurrent MPL and CCLR [12,13,14,15]. Among these, the tibial plateau leveling osteotomy (TPLO) procedure is widely used for treating CCLR [13,16]. Initially introduced by Slocum in 1993, TPLO involves making a circular osteotomy in the proximal portion of the tibia and rotating it to reduce the angle of the tibial plateau, thereby alleviating cranial tibial thrust [12,17]. To correct MPL, procedures such as trochleoplasty and tibial tuberosity transposition (TTT) are commonly used [13,18]. These methods aim to reposition the patella and stabilize the extensor mechanism. TTT, a significant component of MPL surgery, involves making a partial osteotomy of the proximal tibial tuberosity and lateral repositioning of this [13]. These surgical interventions significantly alleviate lameness associated with MPL and CCLR [12]. Combining TTT and TPLO is emerging as a preferred method for correcting both conditions simultaneously [15,16]. However, performing TPLO surgery and trochleoplasty, especially TTT, concurrently in small-breed dogs presents challenges due to anatomical constraints [12,15,16,19,20,21]. Limited space increases the likelihood of interference between screws and pins, often leading to improper placement of the implant and compromised stability [15,19,21]. Additionally, this procedure carries an increased risk of tibial tuberosity fracture or avulsion due to the creation of two osteotomy lines in the proximal tibia [15,19,21]. Performing these surgeries separately poses challenges due to increased risks associated with anesthesia and prolonged recovery times [12,20]. Given that these conditions typically coexist in elderly dogs [3], addressing both issues in a single surgical procedure appears reasonable [12,14,15,22,23].

To mitigate these risks, modified TPLO surgical techniques have been developed to reduce costs and minimize the risks associated with anesthesia in multiple procedures [12,14,15,19,20,21]. This modification involves adapting the standard TPLO plate to enable medial shifting fixation of the proximal tibia, potentially combining the benefits of both TPLO and TTT procedures [12,14,15,22]. However, the process of contouring the plate during surgery may prolong operation time and result in varied outcomes depending on the surgeon’s experience [14]. To address these challenges, several pre-contoured plates and patient-specific implants using 3D printing techniques have been specifically developed for small-breed dogs [9]. During the TPLO procedure, the application of a pre-contoured plate post-osteotomy facilitates medial shifting of the proximal segment, effectively replicating the outcomes of TTT [20]. A novel design of a pre-contoured TPLO plate, angled medially by 2 mm, has been introduced for small-breed dogs, offering advantages such as reduced operation time and increased consistency between surgeons [20]. Despite these advantages, there is currently no confirmation regarding whether contouring affects plate stability and provides adequate medialization in small-breed dogs [12,14]. 

This study aimed to assess and confirm the efficacy of the pre-contoured modified TPLO (PCM–TPLO) plate through a comparative analysis with the traditional TPLO plate group. The study focused on evaluating bone alignment and the capacity for adequate medialization in small-breed dogs, incorporating mechanical testing to assess the stability of the interface between the bone and the plate. The underlying hypothesis suggests that the PCM–TPLO plate not only offers a feasible alternative to TTT but also ensures stability and alignment comparable to that of the traditional TPLO plate. 

## 2. Materials and Methods

### 2.1. Plates

The 1.5/2.0 mm PCM–TPLO plate (Jeil Medical Corp., Seoul, Republic of Korea), made of 316 L stainless steel, was designed to improve alignment and stability in surgical repairs for small-breed dogs (Figure 1). Notably, the newly developed PCM–TPLO plate is engineered with a 2 mm medial inclination, specifically crafted to promote a 2 mm lateral shift of the distal segment upon application (Figure 1D) [24]. The three proximal holes and two distal holes are designed to be engaged with locking screws. The other hole is a dynamic compression hole, providing axial compression when used with a cortical screw.

### 2.2. Specimens

Sixteen stifle joints from eight small-breed dog cadavers were used in this study. All dogs were owned by clients and were euthanized for reasons unrelated to this research, then generously donated to our research team. The cadavers, each weighing between 3 and 5 kg, had no prior history of orthopedic conditions, including MPL or CCLR. A radiographic examination was conducted to confirm the absence of any anatomical abnormalities. Before the procedure, the cadavers were frozen and stored at −20 °C, then thoroughly thawed at room temperature. The 16 stifle joints were randomly divided into two groups: the conventional TPLO plate and PCM–TPLO plate groups, each comprising eight stifles. Care was taken to ensure that the stifle joints from an individual cadaver were evenly distributed between the two groups.

### 2.3. Surgical Procedures

Mediolateral and craniocaudal radiographs were obtained to assess the osteotomy site and identify any anatomical deformities (Figure 2). The same surgical planning approach was maintained for both groups throughout the study by a surgeon (E.B.J). The tibial plateau angle (TPA) was measured in the mediolateral view, following the method described by Slocum and Devine [25], while the mechanical medial proximal tibial angle (mMPTA) was measured in the craniocaudal view using the approach described by Dismukes et al. [26,27]. Specific measurements denoted as D1, D2, and D3 were taken to determine the osteotomy site using the standard method previously described [20].

All subsequent procedures were performed by a single surgeon (E.B.J) based on the earlier surgical planning. The TPLO surgery aimed to achieve a postoperative TPA of 5° [17,24]. The surgeon selected either an 8 or 10 mm saw TPLO blade depending on the size of the cadaver. In cases involving the PCM–TPLO plate, the surgery followed the same method, with an emphasis on maximizing contact between the plate and bone to displace the proximal tibial fragment medially (Figure 2C,D).

### 2.4. Postoperative Measurements

#### 2.4.1. Radiographic Measurements 

Following the surgical procedure, mediolateral and craniocaudal radiographs were obtained following the same protocol. Postoperative measurements, including TPA, mMPTA, the number of screws within the joint, bone–plate gap, medialization distance, and tibial osteotomy width, were then conducted based on these radiographic images and gross assessment (Figure 2 and Figure 3). For cases involving PCM–TPLO, an assessment of the extent of medial translation was performed. This parameter was assessed as the percentage value between medialization distance and tibial osteotomy width.

#### 2.4.2. Compression Test

Following postoperative radiography, we surgically removed all soft tissues and the fibula attached to the tibia. To prevent any interference with the compression test, a 5 mm distal section from the widest part of the tibial tuberosity was excised. An osteotomy was then performed 60 mm distal to the furthest point of the TPLO plate. The mechanical axis of the tibial bone model was aligned perpendicular to the table and securely affixed to a 3D resin jig using plaster (Mungyo Corp., Gimhae, Republic of Korea). This ensured that neither the plate nor the screws were obstructed by the plaster. Similarly, we secured the proximal part of the bone model to prevent the plate and screws from being covered by the plaster. The cadaveric specimens, integrated with the 3D resin jig made from 3D printing UV-sensitive resin (Anycubic, Shenzhen, China), were then positioned in a static load testing machine (Z010 TN; Zwick Roell, Ulm, Germany) equipped with a stainless-steel jig designed to accommodate the 3D resin jig (Figure 4). After applying a preload of 5 N, an axial compressive force was applied at a rate of 2 mm/min until implant failure occurred [28]. Standard force–time curves were generated under these test conditions, with implant failure being defined as the maximum load immediately before a sudden decrease in the continuously applied load [29].

### 2.5. Statistical Analysis

Prior to conducting the study, we performed a priori power analysis using statistical software (G*Power V3.1.9.7) to determine the required number of cadavers [30]. We determined a sample size of four stifle joints based on the following parameters: α = 0.05, power = 0.8, and estimated effect size (d) = 2.5766667, derived from the mean and standard deviation (SD) observed in a pilot study involving four cadavers. However, the final sample included sixteen stifle joints from the pilot study, equally distributed among the two groups. Data were analyzed using SPSS software version 29 (IBM SPSS, Chicago, IL, USA). Additionally, the Kolmogorov–Smirnov test was employed to assess the normal distribution of continuous variables. Differences in data, excluding the number of intra-articular screws, were assessed using an independent *t*-test for each group. The number of intra-articular screws within each group was compared using the Mann–Whitney *U* test. A *p*-value of <0.05 was considered statistically significant.

## 3. Results

### 3.1. Specimens

Sixteen hindlimbs were obtained from eight small-breed canine cadavers collected between March 2023 and September 2023. All cadavers weighed between 3 and 5 kg, including Chihuahua (3), Maltese (3), and mixed breeds (2). During the compression test, two hindlimbs had to be excluded from both the PCM–TPLO plate and the TPLO plate groups due to unintentional intraarticular screw placement. 

### 3.2. Radiographic Measurements

The mean TPA and mMPTA values were measured from eight hindlimbs in each group (Table 1). In both groups, the postoperative TPA appeared to be 5°, as targeted, and no significant differences were found between the two groups (*p* = 0.878). There was also no significant difference in postoperative mMPTA between the two groups, and the changes before and after surgery were considered not statistically significant. However, in the PCM–TPLO plate group, there was a tendency for a decrease in postoperative mMPTA compared with preoperative mMPTA.

The number of screws within the joint was measured. Within the PCM–TPLO plate group, two dogs had one or two screws, respectively, entering the joint capsule (median, 1.5; range: 0–2). In contrast, the TPLO plate group showed no screws entering the joint capsule in any dog, and the difference between the two groups was not significant (*p* = 0.442). 

The mean values of the bone–plate gap for each group are presented in Table 1. In the TPLO plate group, there was virtually no distance between the bone and plate, while the PCM–TPLO plate group exhibited a 1.9 mm gap, indicating a significant difference between the two groups. 

The medialization distance and tibial osteotomy width were exclusively calculated for the PCM–TPLO plate group, demonstrating mean values and standard deviations of 2.275 ± 0.578 mm and 7.674 ± 0.763 mm, respectively. The calculated mean and standard deviation of the degree of medial translation were 0.30 and 0.08. 

Compression tests for load to failure were conducted (Table 1). The plates of both groups were able to withstand forces > 700 N. Notably, no significant differences were observed between the two groups.

## 4. Discussion

In our investigation, we conducted a comparative anatomical and biomechanical analysis using cadaver specimens to examine alignment and compression resistance between the traditional TPLO and PCM–TPLO plates. We evaluated the role of PCM–TPLO plates in mediating surgical outcomes. We found that pre-contouring the plate did not alter postoperative TPA or mMPTA, nor did it increase the incidence of screw penetration into the joint capsule. Although applying the PCM–TPLO plate led to a slightly larger bone–plate gap, this did not compromise the integrity of the plate strength.

TPLO surgery aims to counteract the forward-directed force on the tibia by adjusting the TPA [14,19]. Following surgery, a TPA of 5–6.5° should be maintained to prevent the tibia from moving forward while avoiding backward movement [17,25,31]. Clinically, gait improvement has been reported when the postoperative TPA is within 0–14° [32]. In our study, both the traditional TPLO and PCM–TPLO groups showed mean postoperative TPA values within these recommended parameters, with no marked difference between the two groups. These findings imply that plate contouring does not significantly affect the alignment of the tibial plateau. Therefore, the PCM–TPLO plate is likely to be effective in restoring normal leg function by protecting against both forward and backward thrust of the tibia.

Following TPLO and MPL corrective surgeries, instances of improper bone alignment have been reported, often attributed to suboptimal surgical techniques [24,31,33,34,35]. Misalignments, such as varus or valgus deformities, can be detected by measuring mMPTA [25,36]. According to previous studies, the standard value for mMPTA is around 95.1 ± 3.2° [33,36]. Consistent with this, our study found that both the control (TPLO) and PCM–TPLO plate groups maintained normal mMPTA values before and after surgery. However, a slight increase in postoperative mMPTA was observed in the PCM–TPLO group, contrary to the findings of previous research, but there was no statistical significance [14]. Although there was a minor adjustment, it stayed within the normal limits and showed no significant variance when compared with the standard TPLO plate group. From this, we infer that the PCM–TPLO plate can mediate the medial displacement of the proximal tibial segment while preserving the integrity of the limb’s overall alignment.

The PCM–TPLO plate was designed to enable a medial shift of the tibia by 2 mm. A previous study noted that the application of a pre-contoured T-plate, designed for 2 mm medial displacement on a tibial model, typically resulted in a medial shift typically of 0.05–1 mm [20]. Our findings indicate a median medialization distance of 2.275 ± 0.58 mm, corresponding to approximately 30 ± 8% of the total length of the osteotomy site. The increased medialization observed could be attributed to the unique design of our plate and the effects of the surrounding soft tissue. Previous clinical successes in treatment of MPL grades I, II, and III were reported with an average medialization of 20% [14,20]. Furthermore, for optimal bone healing, the repositioned bone segment should cover at least 50% of the osteotomy gap [8]. Our data, demonstrating approximately 70% coverage, suggest that the PCM–TPLO plate facilitates suitable medialization without compromising bone healing. Nonetheless, further clinical studies are essential to fully ascertain the practical implications of these findings.

In our study, we observed a slightly larger gap between the bone and plate in the PCM–TPLO plate group compared with the standard TPLO plate group. Prior research suggested that a larger gap could decrease stability, raising concerns about the surgery’s success [20]. However, our results revealed that the maximum force the plates could withstand did not significantly differ between groups. Both were capable of supporting average forces greater than 700 N, thereby highlighting the superior mechanical properties of the PCM–TPLO plate compared to those observed in previous studies [28,37]. This discrepancy may be attributable to the differences in plate designs used and methods for measuring force in our study. Considering the typical weight support of a dog’s pelvic limb, which is approximately 20% of its body weight during the convalescent period after surgery, and the maximum force it experiences during activities such as trotting, the PCM–TPLO plate is estimated to handle approximately 70–80% of the dog’s weight [28,38]. The other research indicated that peak force can significantly increase during more vigorous activities [39]. Based on these considerations, for a dog weighing 5 kg, the leg might be subjected to a load of approximately 100 N. Yet, our findings indicate that the PCM–TPLO plate can withstand considerably higher loads than this estimate, even with a 2 mm gap between the bone and the plate. Thus, we conclude that the PCM–TPLO plate remains a reliable and effective option for dogs weighing less than 5 kg, without any increased risk of failure. The adaptation of the PCM–TPLO plate for medialization has raised concerns regarding a potentially higher risk of screw penetration into the joint space. Originally, TPLO plates were designed to prevent the entry of screws into the joint due to the potential for intraarticular screws to inflict meniscal injuries or degenerative changes [40,41]. In the PCM–TPLO group, two screws were observed to penetrate the joint capsule. However, since the PCM–TPLO plate was designed similarly to conventional TPLO plates to prevent screws from intruding into the joint space, such occurrences are likely attributed to the surgeon. Additionally, there was no statistically significant difference between the two groups. Therefore, despite these instances, the use of the PCM–TPLO plate remains a viable surgical option, capable of withstanding significant loads without causing additional damage to the patient.

This study had several limitations. First, a limited number of cadaveric specimens were used. While cadavers provide a closer approximation to the physiological environment, it is important to note that such specimens may differ from the bones of living specimens regarding bone quality. Bones of living animals are typically more robust, and additional anatomical structures contribute to their ability to withstand force, implying that they might endure even higher loads. Additionally, the cadavers used in this study did not have concurrent orthopedic diseases of the stifle joint; thus, the pathological arthrokinematics regarding MPL could not be evaluated. Second, the mechanical test was conducted in only one direction, failing to reflect the combinations of forces from various directions that may occur in real-life scenarios. Because repetitive forces similar to clinical conditions were not assessed, further studies are required to evaluate the practical clinical applications of this technique. Finally, there are limitations to the procedure of lateralizing the tibial tuberosity. High-grade MPL with associated bone deformities, such as femoral varus deformity, decreased femoral anteversion angle, and tibial valgus deformity, cannot be adequately addressed solely by lateralization of the tibial tuberosity to realign the quadriceps muscles. These cases may necessitate additional corrective osteotomies.

## 5. Conclusions

We conducted a comparison between the conventional TPLO group and the PCM–TPLO plate group using cadavers of small dogs weighing between 3 and 5 kg. Our findings from postoperative radiography and mechanical tests confirmed that the PCM–TPLO plate effectively adjusts TPA without increasing the risk of additional complications. Furthermore, owing to its appropriate rearrangement effect of tibial tuberosity related to the extensor mechanism, this innovative plate is expected to emerge as a favorable surgical option for small-breed dogs with concurrent CCLR and MPL.

## Figures and Tables

**Figure 1 animals-14-01937-f001:**
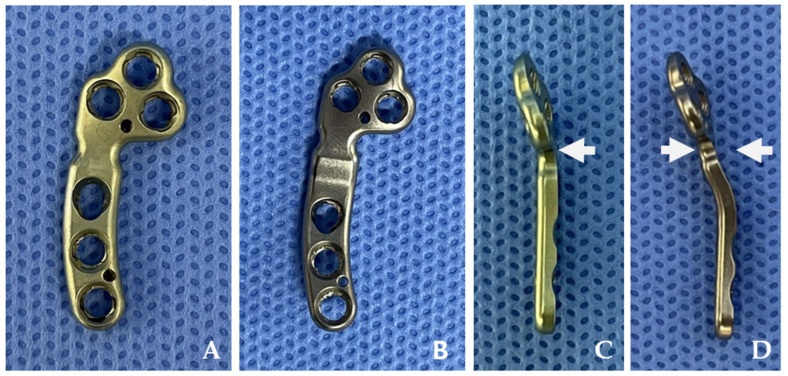
These figures depict a conventional tibial plateau leveling osteotomy (TPLO) plate, showing its frontal view (**A**) and side profile (**C**). The pre-contoured modified TPLO plate is presented in the frontal view (**B**) and side profile (**D**). The pre-contoured plate is specifically shaped to conform closely to the medial aspect of the tibia, intended to induce a relative medial shift of the proximal segment by 2 mm. The white arrows highlight the region of medialization.

**Figure 2 animals-14-01937-f002:**
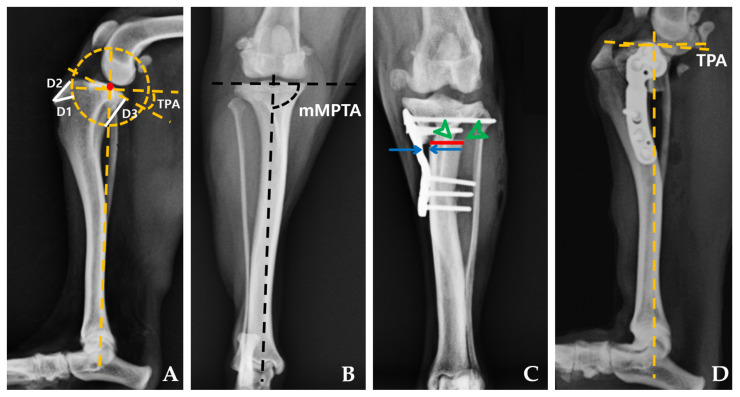
Pre- and postoperative mediolateral and craniocaudal radiographs for TPLO. Preoperative mediolateral radiographs illustrating measurements of the tibial plateau angle (TPA), D1 is the distance from the perpendicular cranial straight edge of the tibial crest at the most cranio-proximal point of the tibial tuberosity to the intended osteotomy site. D2 extends from the most cranio-proximal point of the tibial tuberosity to where the intended tibial osteotomy intersects the cranial tibial subchondral bone. D3 measures from the subchondral bone at the most caudal margin of the tibial plateau to where the intended tibial osteotomy intersects the caudal tibial cortex (**A**). The preoperative measurement of mMPTA is depicted in (**B**). Postoperative radiographic measurements conducted on craniocaudal radiographs for the PCM–TPLO group (**C**) include tibial osteotomy width (red line), bone–plate gap (blue arrow), and medialization distance (green arrowhead). The postoperative measurement of TPA is shown in (**D**).

**Figure 3 animals-14-01937-f003:**
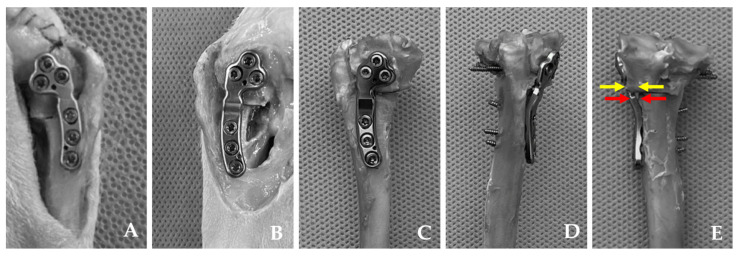
Intraoperative images illustrating the attachment of the TPLO plate (**A**) and the pre-contoured modified (PCM) TPLO plate (**B**) to the tibia. Measurements of the dissected tibiae of the PCM–TPLO groups (**C**–**E**). Medial, anterior, and posterior views after removal of soft tissues and the fibula attached to the tibia (**C**–**E**). The extent of medial translation of the proximal segment was measured using calipers (yellow arrows), while the gaps between the bone and plate were measured (red arrows).

**Figure 4 animals-14-01937-f004:**
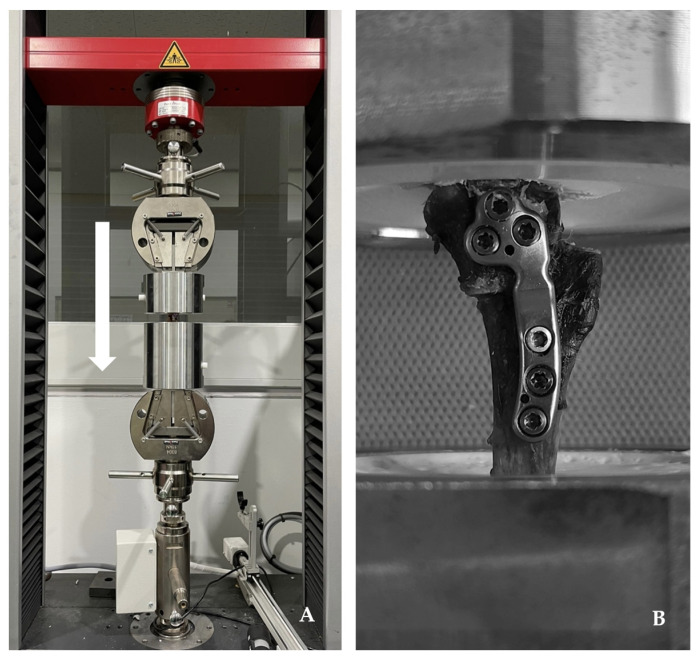
Photograph of the compression test. Mechanical testing stainless-steel jig affixed to the servo-hydraulic testing machine. A 3D resin jig is firmly positioned within a stainless-steel jig, and a compressive load is applied vertically downward (**A**). The tibia with the TPLO plate is securely fastened to the proximal and distal 3D resin jigs, ensuring that neither the plate nor the screws are embedded in plaster (**B**).

**Table 1 animals-14-01937-t001:** Descriptive data for the TPLO and PCM–TPLO plate groups (mean ± SD).

Parameters	TPLO Plate Group		PCM–TPLO Plate Group		*p* Value
	Mean	SD	Mean	SD	
Preoperative TPA (°)	25.00	2.00	25.25	2.86	0.798
Postoperative TPA (°)	5.48	1.66	5.38	2.15	0.878
Preoperative mMPTA (°)	96.96	4.33	96.13	6.17	0.721
Postoperative mMPTA (°)	97.04	1.84	94.34	4.38	0.131
Bone–plate gap (mm)	0.59	0.54	1.90	0.44	0.001
Load to failure (N)	798.80	375.02	721.50	292.02	0.818

TPLO: tibial plateau leveling osteotomy, PCM–TPLO: pre-contoured tibial plateau leveling osteotomy, TPA: tibial plateau angle, mMPTA: mechanical medial proximal tibial angle.

## Data Availability

The original contributions presented in the study are included in the article, further inquiries can be directed to the corresponding author/s.

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
