# Peer review of "Assessing the Effectiveness of Modified Tibial Plateau Leveling Osteotomy Plates for Treating Cranial Cruciate Ligament Rupture and Medial Patellar Luxation in Small-Breed Dogs"

_animals, 2024, doi:10.3390/ani14131937_

Round 1

Reviewer 1 Report

Comments and Suggestions for Authors
  • A brief summary 

In small-breed dogs, medial patellar luxation and cranial cruciate ligament rupture often co-occur, presenting through various mechanisms. A pre-contoured modified tibial plateau leveling osteotomy plate (PCM-TPLO) was designed to achieve both tibial tuberosity transposition and tibial plateau angle reduction. This cadaveric study examined the efficacy and stability of the modified plate. Findings indicate that the PCM-TPLO effectively prevents tibial forward displacement and medial patellar luxartion while maintaining stability.

  • General concept comments

The general concept of the paper is sound. The only methodological accuracy found was that the testing setup description did not match the photographs. The authors state that the specimens were prepared in a way to prevent implant interference with the resin used to form testing jig attachments, but the image in Figure 4 shows the distal part of the plate imbedded in the resin. Was this an omssion, a mistake, maybe a result of perspective of the photo taken or something else?

  • The manuscript is clear, relevant to the field and presented in a well-structured manner. However, it should be noted that for grade III MPLs corrective osteotomies of the distal femur and proximal tibia are  gaining traction and clinical relevance and should be mentioned in the discussion. In this form, the Discussion leaves the reader with the impression that medialization of the proximal osteotomy during TPLO is a comprehensive and overall successful treatment method for grade III MPLs.   
  • The references (1), (4), (8) and (13) are somewhat outdated and should be replaced with newer ones or omitted.

The manuscript is scientifically sound and is the experimental design is appropriate to test the hypothesis. The design of the testing jig described by Kowaleski et al. [Kowaleski MP, Apelt D, Mattoon JS, Litsky AS. The effect of tibial plateau leveling osteotomy position on cranial tibial subluxation: an in vitro study. Vet Surg. 2005 Jul-Aug;34(4):332-6. doi: 10.1111/j.1532-950X.2005.00051.x. PMID: 16212587] would be more appropriate because this testing setup is more representative of in vivo biomechanics of the canine stifle.

  • The manuscript’s results are reproducible based on the details described in the methods section.
  • Figure 4 (B) shows the test specimen fastened to the proximal and distal 3D resin jigs, but the photograph chosen shows the distal part of the TPLO plate supported and possibly incorporated by the resin base. Is this due to perspective or superposition? If possible, choose a different photograph. If this is not possible, please change the description to correspond to the figure.
  • The conclusions are mostly consistent with the evidence and arguments presented. However, since the modified TPLO plate was tested in vitro without a complete patellar and quadriceps mechanism, and the cadaveric group did not consist of patients with deformities which cause patellar luxation, additional research in vivo with MPL grade II or III inclusion criteria should be done first before claiming the expectancy of this implant emerging as a favorable surgical option. This may well be possible, but the results of this paper suggest only the possibility and mechanical feasibility of such an option for tibial mechanical stability, not for knee arthrokinematics.
  • Ethics approval for the paper is lacking. Without it, this paper cannot be published.
  • P5L171: To prevent any interference with the compression test, we excised a 5-mm distal portion from the widest part of the tibial tuberosity was excised. Please fix this sentence.
  • P5L178: The bone model, integrated with the 3D resin jig made from 3D printing... This is a cadaveric specimen, not a model.
  • Informed Consent Statement: Written informed consent has been obtained from the owner of the 343 patient to publish this paper. The actual cat remained anonymous. This statement has to be changed as it has nothing to do with the paper.
Comments on the Quality of English Language

The language and phrasing in some sentences should be improved. 

Reviewer 2 Report

Comments and Suggestions for Authors

Comments and suggestions for the authors

Thank you for submitting this interesting article to Animals, which aimed to evaluate and confirm the efficacy of modified precontoured TPLO plate (PCM-TPLO) compared to conventional TPLO plate.

Overall, the quality of the English is good and no linguistic revision is necessary. The study is very interesting, well structured and well written. In my opinion, after appropriate editing, its publication could enrich the existing bibliography. 

Here are my comments.

Lines 49-56: In the introductory section, the etiopathogenesis of medial patellar luxation is mentioned. In my opinion, it would be better to speak of patellar luxation in general because the etiopathogenetic mechanism underlying this condition can generate both medial and lateral luxation, depending on the size of the animal (small dogs/large dogs) and other factors.

Furthermore, although the pathogenesis remains unclear, PL is often associated with malalignment of the quadriceps muscles, patella and patellar tendon, as mentioned. However, bony deformities (femoral and tibial) are another key factor in the etiopathogenetic mechanism of this condition, as is misalignment of the extensor (or quadriceps) mechanism of the shinbone, consisting of the quadriceps muscle group, patella, patellar tendon and ligament, trochlear groove and tibial tubercle. Bone deformities are complex musculoskeletal disorders involving varying degrees of PL during growth. I suggest expanding the bibliography in this regard (example: Panichi, E.; Cappellari, F.; Burkhan, E.; Principato, G.; Currenti, M.; Tabbì, M.; Macrì, F. Patient-Specific 3D-Printed Osteotomy Guides and Titanium Plates for Distal Femoral Deformities in Dogs with Lateral Patellar Luxation. Animals 2024, 14, 951. https://doi.org/10.3390/ani14060951)

Lines 78-92: With regard to the section describing the risks associated with the intraoperative plate shaping process, I would add to the development of pre-packaged plates specifically for small dogs a mention and bibliography on other possible options. For example, modern 3D (three-dimensional) printing techniques allow the production of patient-specific instruments (PSI) (example: Panichi, E.; Cappellari, F.; Burkhan, E.; Principato, G.; Currenti, M.; Tabbì, M.; Macrì, F. Patient-Specific 3D-Printed Osteotomy Guides and Titanium Plates for Distal Femoral Deformities in Dogs with Lateral Patellar Luxation. Animals 2024, 14, 951. https://doi.org/10.3390/ani14060951).

The Materials and Methods section is accurately described. The only question I have is why was it not considered to use an advanced diagnostic method such as CT for pre- and postoperative angle measurement?

Section 2.4.2. 'Compression test

Line 171-172: correct this sentence "To prevent any interference with the compression test, we excised a 5-mm distal portion from the widest part of the tibial tuberosity was excised." for example "To prevent any interference with the compression test, a 5 mm distal section was excised from the widest part of the tibial tuberosity."

The discussions are well argued.

Reviewer 3 Report

Comments and Suggestions for Authors

Comments and Suggestions for Authors:

Congratulations on your research. The paper is well done. I have a few suggestions to improve the clarity of the text.

Lines 13-14: Remove "(kneecap dislocation)" and “(knee ligament tears)” from the sentence. These definitions do not contribute anything.

Line 14: Replace “older age” with “adult dog”. Using "older age" gives the impression that the affected dogs must be very old, whereas the concomitant pathology normally occurs in adult dogs, but they are not necessarily old.

Line 17: Do not use abbreviations in the Summary. Delete “(PCM-TPLO)”

Line 20: Replace “the PCM-TPLO” with “this pre-contoured plate”

Line 30: Replace “TPLO” with “tibial plateau leveling osteotomy (TPLO)”

Lines 59-60: Replace “TPLO involves reshaping the tibia into a cylindrical form and rotating it to reduce the angle of the tibial plateau” for “TPLO involves making a circular osteotomy in the proximal portion of the tibia and rotating it to reduce the angle of the tibial plateau”.

Lines 64-65: Replace “TTT, a significant component of MPL surgery, involves the partial excision and lateral repositioning of the tibial tuberosity” for “TTT, a significant component of MPL surgery, involves making a partial osteotomy of the proximal tibial tuberosity and lateral repositioning of this”

Lines 102-106: The authors must indicate whether the plate used is a locking plate or not. They should justify in the discussion the use or not-use of this type of implant for the described technique. My recommendation is that, taking into account the characteristics of the procedure, the plate should be locked with a Locking Compression Plate (LCP) System.

Line 133: The Quote number 19 is “Slocum and Slocum”. If the quote is “Slocum and Devine” it should be number 32.

Line 135: Specific measurements D1, D2, and D3 are known to surgeons performing TPLO, but they should be indicated at this point or in the figure legend, specifying what each measurement corresponds to, so readers do not have to refer to the original article.

Lines 148-149: Replace “(C) include tibial osteotomy width (black line), bone-plate gap (black arrow), and medialization distance (arrow- head)” for “(C) include tibial osteotomy width (red line), bone-plate gap (yelow arrow), and medialization distance (green arrow- head)”. The x-ray is black and white, so colored markers will stand out better against this background.

Lines 165-166: I think the colored markers will stand out better on a black and white background.

Line 344: I do not understand the phrase: “The actual cat remained anonymous.”

Round 2

Reviewer 1 Report

Comments and Suggestions for Authors

The general concept of the paper is sound.

The manuscript is clear, relevant to the field and presented in a well-structured manner.  Discussion has been edited per request.

·         The references are relevant, contemporary and contain no self-citations.

·         The manuscript is scientifically sound and is the experimental design is appropriate to test the hypothesis.

  • The manuscript’s results are reproducible based on the details described in the methods section.
  • Figure 4 (B) has been fixed.
  • The conclusions are mostly consistent with the evidence and arguments presented. This section has been edited per request and is in line with the results of the article.
  • Ethics approval for the paper is lacking. Institutional Review board statement was put in it's place. A corresponding register number of the Review Board's opinion or statement should be included.
  • P5L178: To prevent any interference with the compression test, we excised a 5-mm distal portion from the widest part of the tibial tuberosity was excised. Please fix this sentence.
Comments on the Quality of English Language

Dear Editor,

The article has been revised in accordance to previous comments.

Minor English editing is needed, i.e.

P5L178: To prevent any interference with the compression test, we excised a 5-mm distal portion from the widest part of the tibial tuberosity was excised.